

# Chloropid flies (Diptera, Chloropidae) associated with pitcher plants in North America

Julia J. Mlynarek[1,2] and Terry A. Wheeler[2,†]

[1] Harrow Research and Development Centre, Agriculture and Agri-Food Canada, Harrow, Ontario, Canada
[2] Department of Natural Resource Sciences, McGill University, Sainte-Anne-De-Bellevue, Quebec, Canada
[†] Deceased.

## ABSTRACT

We review the taxonomy and ecology of Chloropidae (Diptera) associated with pitcher plants (Sarraceniaceae) in North America. *Tricimba wheeleri* Mlynarek sp.n. is described from the pitchers of *Sarracenia alata* Alph.Wood and *S. leucophylla* Raf. in the southeastern United States (Alabama, Mississippi). *Aphanotrigonum darlingtoniae* (Jones) associated with *Darlingtonia californica* Torr. in northern California is redescribed, including the first description of male genitalic characters. A lectotype is designated for *A. darlingtoniae*. Published records of other species of *Tricimba* Lioy in pitcher plants in North America are considered accidental or facultative occurrences; published records of *Aphanotrigonum* Duda as pitcher plant associates in eastern North America are probably errors in identification.

## INTRODUCTION

There are two very different ways in which insects can associate with pitcher plants (Sarraceniaceae): they can die in the modified, pitcher-shaped leaves and become a source of nutrients for the plant; or they can live in the pitchers and become active partners in a fascinating ecosystem. Several families of Diptera in North America include pitcher plant associated species: ten species of Sarcophagidae (*Dahlem & Naczi, 2006*) and one or two species each of Culicidae, Chironomidae, Sciaridae, and Chloropidae (*Jones, 1916*; *Jones, 1920*; *Szerlip, 1975*; *Folkerts, 1999*; *Dahlem & Naczi, 2006*).

*Jones (1916)* described a new species of Chloropidae, *Botanobia darlingtoniae* Jones (transferred to *Aphanotrigonum* Duda by *Sabrosky (1965)*) from specimens reared from pitchers of *Darlingtonia californica* Torr. in California, but that fly has not been documented in association with pitcher plants since its original description. There are records of a second, undescribed species of *Aphanotrigonum* Duda from multiple species of *Sarracenia* L. in eastern North America (*Folkerts, 1999*). However, based on examination of specimens collected from *Sarracenia* spp. in eastern North America, the eastern chloropids are not congeneric with *A. darlingtoniae* but actually belong to *Tricimba* Lioy, an externally similar, but distantly related, genus.

Corresponding author
Julia J. Mlynarek,
julia.mlynarek@gmail.com

In this paper, we revise the two species of Chloropidae known to be associated with pitcher plants in North America, with a redescription of *A. darlingtoniae* from the western United States and a description of *Tricimba wheeleri* Mlynarek **sp. n.** from the southeastern United States. Adults of *T. wheeleri* have unusual modifications of the tarsi not known in other species of *Tricimba*, possibly associated with locomotion by adult flies inside pitchers of their host plants.

## METHODS

Specimens studied are deposited in the following collections: Lyman Entomological Museum, McGill University, Ste-Anne-de-Bellevue, Quebec, Canada (LEMQ); Personal collection of Robert F.C. Naczi (RFCN); United States National Museum of Natural History, Smithsonian Institution, Washington, D.C., USA (USNM).

Two specimens of *A. darlingtoniae* from LEMQ were submitted to the Canadian Centre for DNA Barcoding (University of Guelph, Guelph, Ontario, Canada) for sequencing of the 658 bp DNA barcode region of the mitochondrial gene CO1.

Fieldwork in California was facilitated by a California State Park Scientific Collecting Permit (2009) number 9-0735 issued to SD Gaimari (CSCA).

Male genitalia were prepared by removing the abdomen of pinned specimens and clearing them in 85% lactic acid heated in a microwave oven for 1–2 periods of 10 s, separated by a 1 min cooling period.

Morphological terms follow *Cumming & Wood (2009)*.

The electronic version of this article in Portable Document Format (PDF) will represent a published work according to the International Commission on Zoological Nomenclature (ICZN), and hence the new names contained in the electronic version are effectively published under that Code from the electronic edition alone. This published work and the nomenclatural acts it contains have been registered in ZooBank, the online registration system for the ICZN. The ZooBank LSIDs (Life Science Identifiers) can be resolved and the associated information viewed through any standard web browser by appending the LSID to the prefix http://zoobank.org/. The LSID for this publication is: urn:lsid:zoobank.org:pub:67933A14-78D1-4A3B-AC17-B00951574F39. The online version of this work is archived and available from the following digital repositories: PeerJ, PubMed Central and CLOCKSS.

## RESULTS AND DISCUSSION

### *Aphanotrigonum darlingtoniae* (Jones)

*Botanobia darlingtoniae Jones, 1916*: 389. Type locality: Mount Eddy, near Sisson, Siskiyou County, California (see Comments).

*Aphanotrigonum darlingtoniae*: *Sabrosky, 1965*: 785.

Type material. Lectotype (here designated): M labelled: "Sisson, Cal./Bred. F. M. Jones./VIII-5-1915"; "m# TYPE"; "Type No./20318/U.S.N.M."; ("Oscinella/darlingto-niae/Type.) Jones" (USNM).

Paralectotypes F: same data as lectotype except "VIII-15-1915"; "f# TYPE"; "Allotype No./20318/U.S.N.M." (USNM). F: same except: "VII-29-1916" [probably error for 1915]; 'Botanobia/darlingtoniae/Type] Jones"; red square blank label] (USNM) (See Remarks).

Other material examined. USA: CA: Siskiyou Co. 6 km NW Mt Eddy, Forest Road 17 (41.3573°, −122.5409°), 2,060 m, sweep wet *Darlingtonia* meadow, 08.vi.2009. J Mlynarek (5M, 6F, LEMQ), same data except TA Wheeler (3M, 1F, LEMQ).

Diagnosis. (Figs. 1– 2) *Aphanotrigonum darlingtoniae* can be distinguished from the other described Nearctic species of the genus (*A. scabrum* (Aldrich), *A. trilineatum* (Meigen)) by the combination of: dark bristles on gena, three brown stripes on scutum in the shape of a lyre, sides of scutellum also brown (uniformly gray in paratypes of *scabrum* from Treesbank), interfrontal setulae are darker, more contrasting with frons in *darlingtoniae* and distinction between gray triangle and yellow anterior part of frons seems less clear in *darlingtoniae* than in *scabrum*.

Description (Fig. 1). Total length 2.3–3.1 mm . Overall colour black, frontal triangle brown-black, pollinose, wide, 2/3 length of frons; ocellar tubercle black, pollinose; frons yellow to brown, wider than long; cephalic setae dark, strong but short, frontal-orbital bristles short, black and convergent, two rows of interfrontal setulae inside the margin of frontal triangle; gena wide, anterodorsally yellow, ventrally black, 0.3–0.4 times eye height; eye bare; postgena black, pubescent and 1/5 width of eye; face yellow to brown; scape and pedicel yellow, first flagellomere oval, 1.25–1.35 higher than long, first segment of arista black, rest of arista black, slender and 2–2.5 times width of first flagellomere; palpus yellow, proboscis brown, clypeus black and shiny.

Scutum black with three brown dorsal lines, grey pollinosity, 1.2–1.4 times wider than long, pronotum black, shiny, thin; scutellum black, pollinose, 1.4–1.5 times as wide as long, apical scutellar bristles black and stronger than surrounding setae, on very small tubercles; thoracic pleura pollinose; dorsal margin of anepimeron and lateral region of postscutellum pollinose. Femur black, tibiae yellow with thick black band, metatarsi yellow; femoral organ absent; tibial organ present, black, 0.3 length of tibia. Ratio of costal sectors C1:C2:C3:C4 on wing 1:1.3:1:0.5.

Abdomen black, pollinose, longer than thorax; syntergite 1 + 2 same length as remaining tergites.

Male postabdomen (Fig. 2): small; epandrium in lateral view 1.4 times higher than wide, in posterior view 1.5 times wider than high, setae sparse covering posterior portion of epandrium; surstylus 0.75 times as high as epandrium, parallel sided with a rounded tip, setae short and sparse; cercus square in posterior view, triangular in lateral view, separated by a small evenly rounded arc, bristles short with one long bristle on the posteromedial edge; distiphallus not well sclerotized.

Molecular sequence data. DNA barcodes (658 bp of CO1) for two LEMQ specimens of *A. darlingtoniae* are available on the Barcode of Life Database (boldsystems.org), (BOLD accession numbers: CCDB-21328-A12/LYMAA1247-14; CCDB-21328-B01/LYMAA1248-14).

**Remarks.** *Jones (1916)* recorded the type locality as "Mount Eddy, near Sisson, Siskiyou County, California". Sisson is the former name for the town of Mount Shasta. There

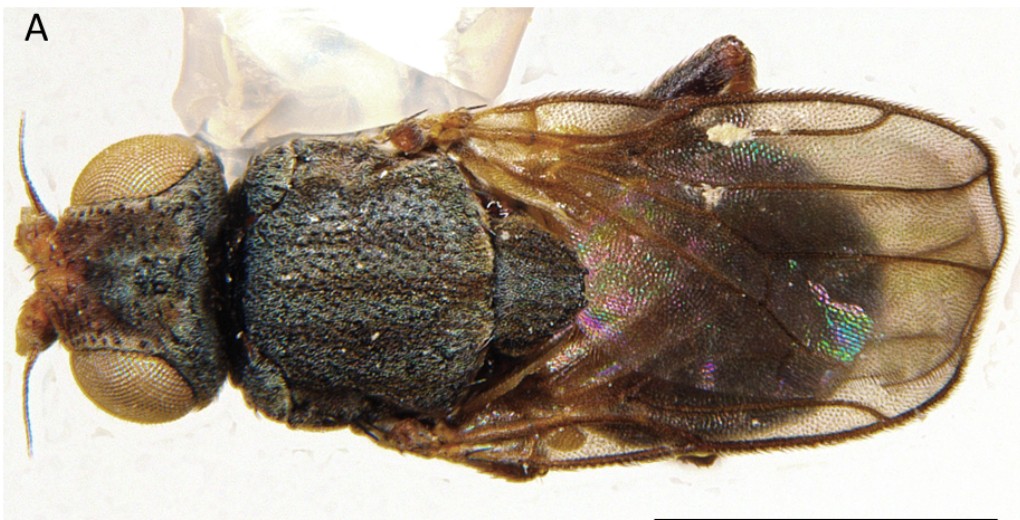

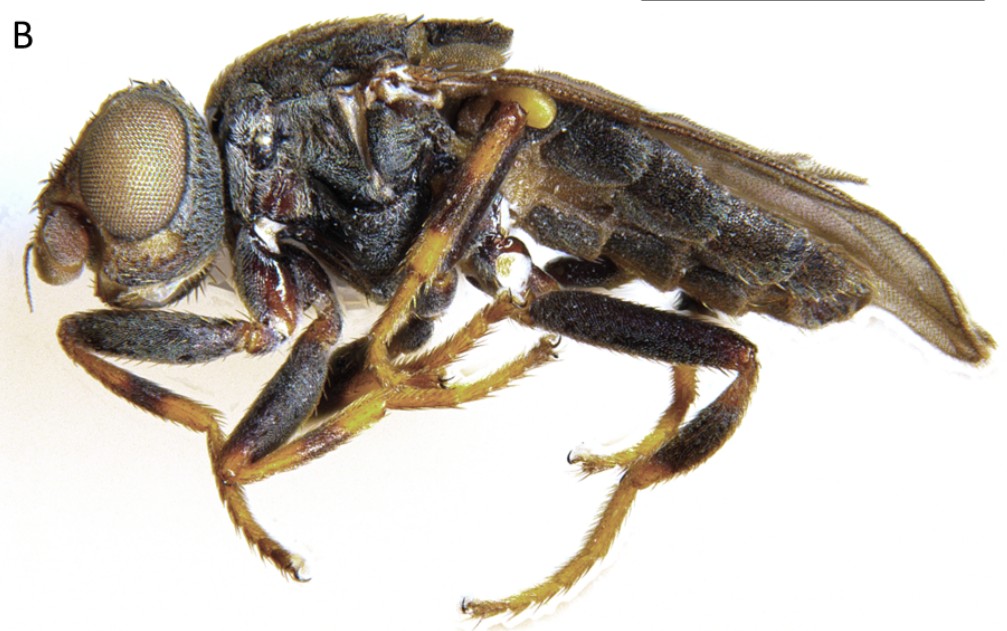

**Figure 1** *Aphanotrigonum darlingtoniae,* **male habitus.** (A) Dorsal. (B) Lateral Scale bar = 1 mm (photos by J Mlynarek).

are several populations of *Darlingtonia californica* in the area surrounding Mount Eddy, west of Mount Shasta, so it is impossible to determine precisely the type locality. The second female type specimen is not labelled as a type either but is assumed and now designated as a paralectotype, because the emergence data matches that given in the original publication. In addition, the determination label is in Jones' handwriting.

Our 2009 collections occurred early in the season (Fig. 3). *Darlingtonia* pitchers were small and pale green, in contrast to the previous year's dark pitchers. There were still patches of snow in the open forest near the collecting site and other herbaceous plants

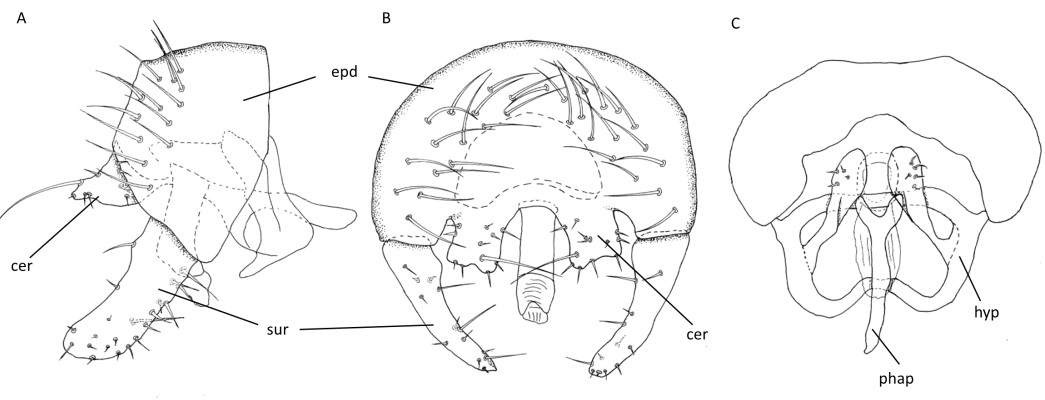

**Figure 2** *Aphanotrigonum darlingtoniae*, **male genitalia.** (A) Lateral. (B) Posterior. (C) Ventral. Abbreviations: cer, cerci; epd, epandrium; hyp, hypandrium; phap, phallapodeme; sur, surstylus. Scale bar = 0.1 mm (drawings by J Mlynarek).

in the vicinity were early in their annual development. Sampling was non-destructive; we swept just above the substrate adjacent to the developing pitchers and over the debris of the previous year's growth. *Jones (1916)* based his description of *A. darlingtoninae* on adult specimens reared from pitchers of *D. californica*. Our collecting supports *Jones*' (*1916*) suggestion that adults can live outside pitchers and, given the small size of new pitchers, the adult flies may overwinter in the substrate outside the pitchers. *Jones (1916)* also described the immature stages of *A. darlingtoniae,* which were subsequently reviewed by *Johannsen (1935)*.

Our collected specimens of *A. darlingtoniae* from California clusters in the same BIN as *Aphanotrigonum trilineatum* from Alberta, British Columbia, and New Brunswick. We morphologically compared the two species to confirm the validity of *A. darlingtoniae*. We also dissected and compared male genitalia. Even though they share the same BIN and molecular barcode, *Aphanotrigonum darlingtoniae* is morphologically distinct from *A. trilineatum* supporting the validity of both species.

### *Tricimba wheeleri* Mlynarek new species
LSID: urn:lsid:zoobank.org:act:C9CC48AD-AE7A-49DB-B849-2C64B6C5E812
Type locality. USA: Alabama: Mobile County, 5mi W of Citronelle.
Type material. Holotype M: USA: Alabama: Mobile County, 5mi W of Citronelle, 31.08° N 88.3°W 2.viii.1994, RFC Naczi, Ex. *Sarracenia leucophylla* (LEMQ). Paratypes: 24 M, 31 F: same as holotype; 1 F: Theodore, 21.vi.1915, FM Jones (USNM); Mississippi: 14 M, 22 F: Stone County, 9mi E of Wiggins, 30.8°N 88.9°W 1.viii.1994, RFC Naczi, Ex *Sarracenia alata.* (LEMQ).
Other material (not examined). USA. Alabama: Baldwin Co., 0.9 mi NNE of Perdido, 31.0206°N, 87.6234°W, 28 April 1988; RFC Naczi, adults collected from inside upper portions of *Sarracenia leucophylla* pitchers. (2M 3F, RFCN) Florida: Okaloosa Co., 3 mi S of Crestview, 30.6990°N, 86.5745°W, 19 Aug. 1984; RFC Naczi, adults collected from inside upper portions of *Sarracenia leucophylla* pitchers. (3M, RFCN).

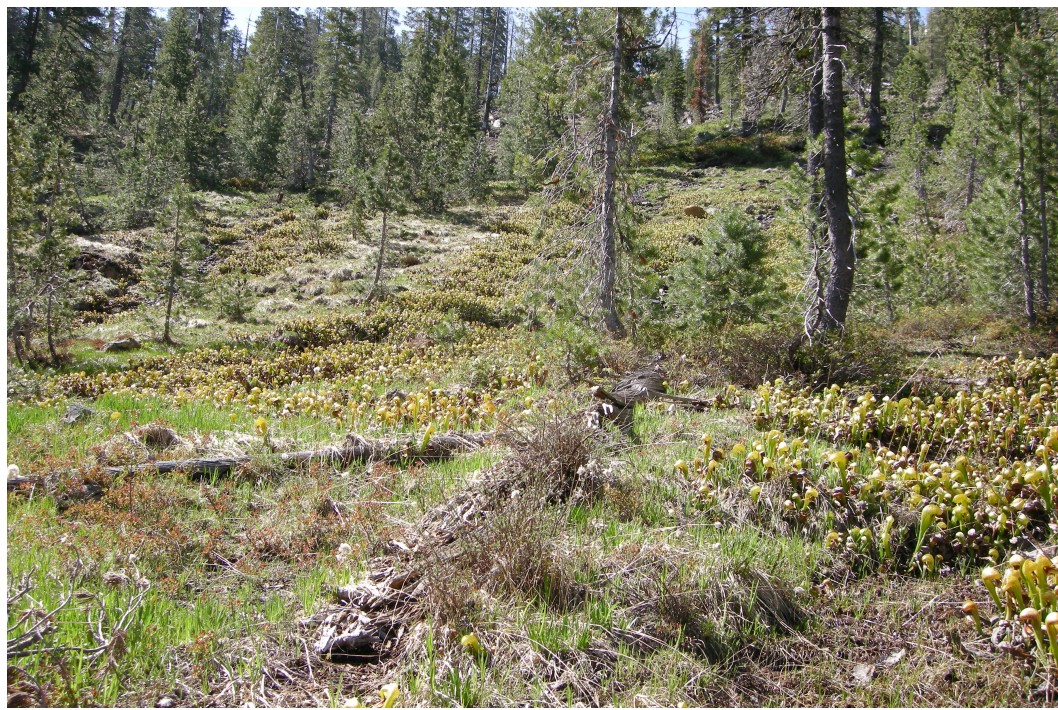

**Figure 3** **2009 collecting locality of *Aphanotrigonum darlingtoniae*. Mt Eddy, Siskiyou County, California.** (photo by T Wheeler).

Diagnosis (Figs. 4–6). *Tricimba wheeleri* can be distinguished from the other described Nearctic species of the genus by the combination of: shallowly incised scutal lines, and heavily microtomentose gray scutum, expanded distal tarsal segments (Fig. 6).

Description (Figs. 4 and 6). Total length 1.7–2.7 mm. Overall colour brown to black, frontal triangle black, pollinose, 0.4–0.5 times length of frons; ocellar tubercle black, pollinose; frons yellow anteriorly, darkening posteriorly until black, size 1.2–1.3 times longer than wide; cephalic setae slender and pale, frontal-orbital bristles even, interfrontal setulae on margin of frontal triangle; gena yellow, pollinose, 0.2–0.3 times eye height; eye bare; postgena black, pollinose and thin; face brown, as high as wide; scape and pedicel yellow, first flagellomere yellow proximoventrally darkening distodorsally, negligibly higher than wide, first segment of arista yellow to brown, rest of arista slender, brown, sparsely hairy and twice as long as first antennal segment; palpus and proboscis yellow, clypeus brown.

Scutum black, pollinose, with three weakly incised dorsal lines along the dorsocentral and acrostichal lines, covered in fine yellow hairs, 1–1.2 times longer than wide, pronotum black and shiny; postpronotum brown to black; scutellum yellow, round, 1.6–1.8 times as wide as long, apical scutellar bristles black and a lot longer than surrounding setae, on small tubercles; thoracic pleurites brown to black, pollinose; dorsal margin of anepimeron and lateral region of postscutellum pollinose. Legs brown, joints sometimes paler; femoral organ absent; tibial organ present, yellow to black, large, 0.3 length of tibia.

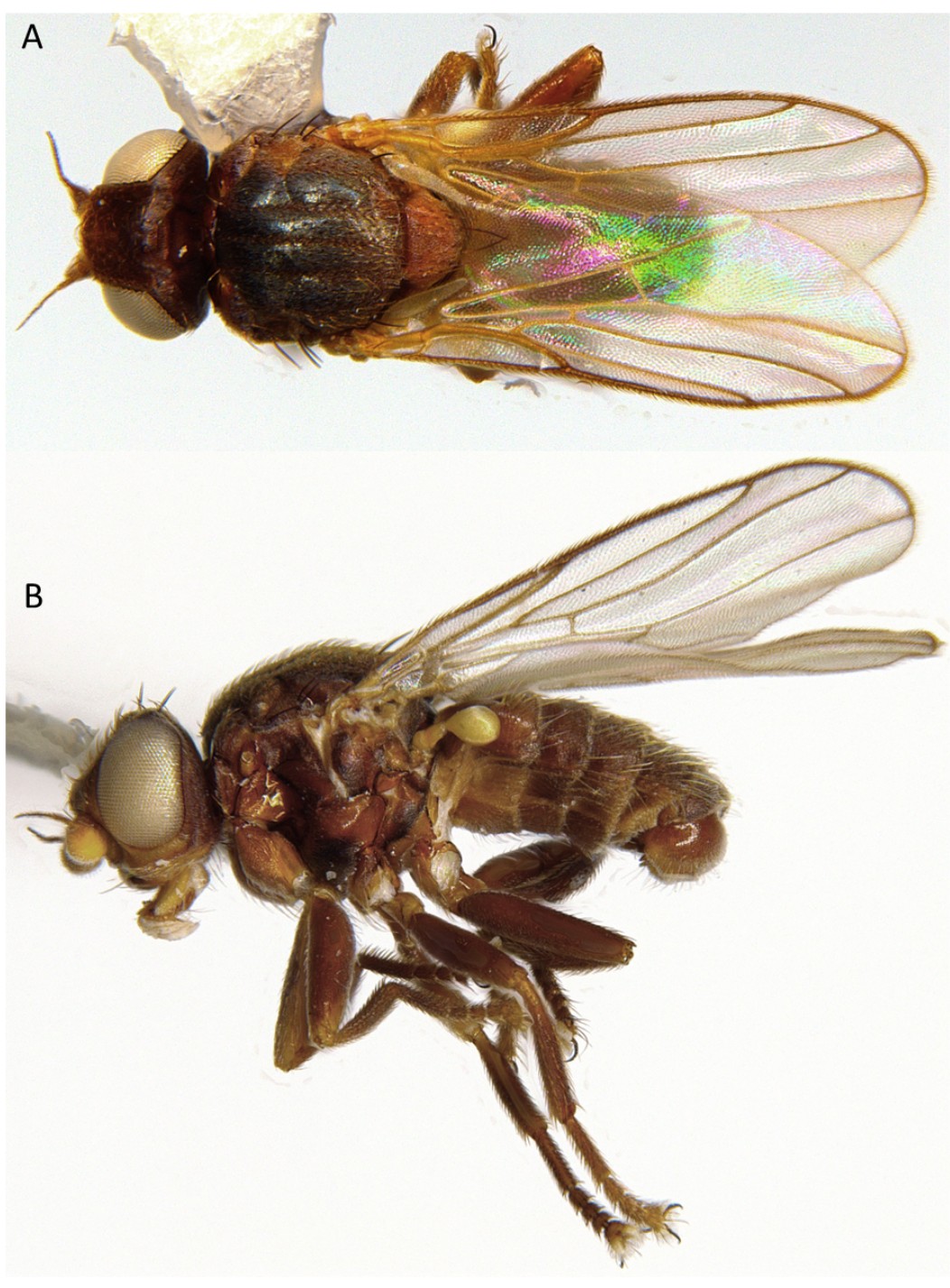

**Figure 4** ***Tricimba wheeleri*, male habitus.** (A) Dorsal. (B) Lateral. Scale bar = 1 mm (photos by J Mlynarek).

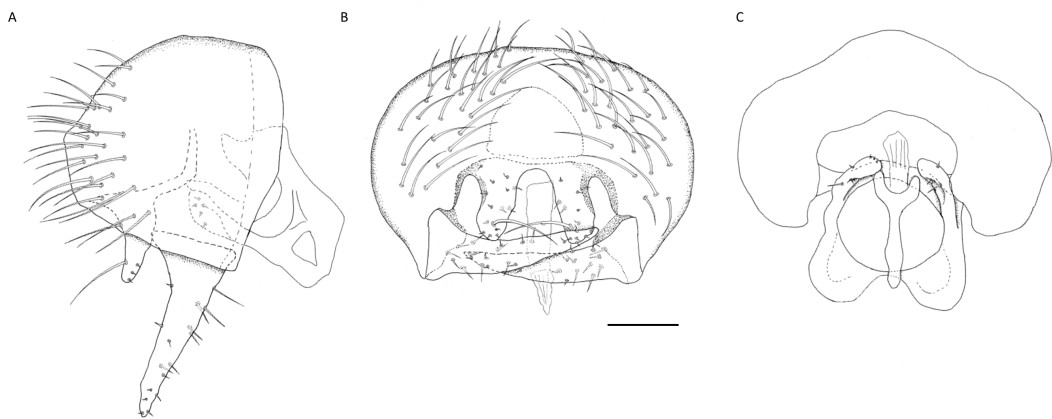

**Figure 5** *Tricimba wheeleri,* **male genitalia.** (A) Lateral. (B) Posterior. (C) Ventral. Scale bar = 0.1 mm (drawings by J Mlynarek).

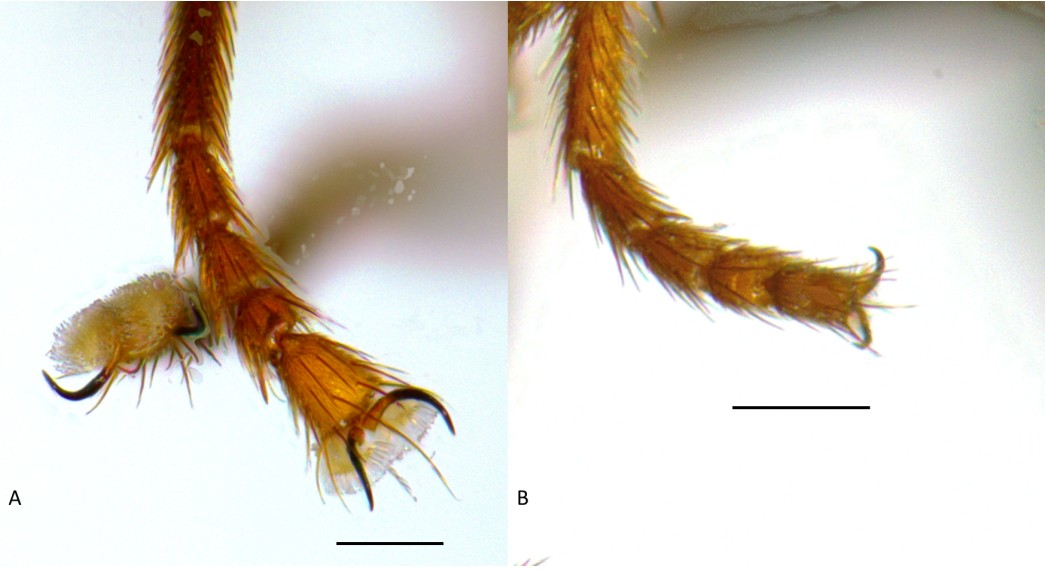

**Figure 6** (A) *Tricimba wheeleri,* **male fore tarsus.** (B) *Tricimba melancholica,* **male fore tarsus.** Scale bars = 0.1 mm (photos by J Mlynarek).

Distal tarsal segments expanded, pulvilli enlarged and long tarsal claws. Ratio of costal sectors C1:C2:C3:C4 on wing 1:0.95:0.6:0.5.

Abdomen black, pollinose, longer than thorax; syntergite 1+2 same length as remaining tergites.

Male postabdomen (Fig. 5): large; epandrium in lateral view as high as wide, in posterior view 1.4 times as wide as high, setae densely covering posterior end of epandrium; surstylus 0.8 times height of epandrium, setae short and sparsely covering entire surstylus; cercus large, parallel sided and rounded apically, separated by small evenly rounded arc, bristles short, covering entire cercus; distiphallus small and poorly sclerotized.

Etymology. The species name is a genitive patronym in honour of Terry A. Wheeler, the co-author of this manuscript who passed away during the final stages of this project and contributed much to our knowledge of the Chloropidae. He always said that "this is probably the last species of insect that is associated with North American pitcher plants that needs to be described, how fitting it should be a chloropid".

Remarks. In addition to collecting a single paratype, FM Jones made notes on additional specimens from Alabama, North Carolina, and South Carolina apparently belonging to this species. Unpublished sketches in the FM Jones archives, deposited in the Peabody Museum of Natural History, Yale University, New Haven, CT, USA (http://harvardforest. fas.harvard.edu/botanobia-darlingtoniae), of specimens collected from *Sarracenia drummondii* (= *S. leucophylla*), *S. flava*, *S. minor* and *S. rubra* illustrate the expanded distal tarsal segments, long tarsal claws, and expanded pulvilli.

The type specimens from *Sarracenia leucophylla* and *S. alata* were collected as adults from inside the upper portion of the pitchers (RFC Naczi, pers. comm., 2000).

The tarsal modifications are not present in other Nearctic *Tricimba* species and may be an adaptation for walking on the surface of the debris inside pitchers (Fig. 6).

*Folkerts (1999)* recorded an undescribed species of "*Aphanotrigonum*" from *Sarracenia* species in the eastern United States and that record has been repeated by others (e.g., *Dahlem & Naczi, 2006*). Although we have not seen vouchers of the specimens referred to by Folkerts, that record likely refers to *Tricimba wheeleri*. *Folkerts (1999)* named, but never published, the species from *Sarracenia* species in the eastern United States as the sister species to *Aphanotrigonum darlingtoniae*; however, we do not consider it the description as a valid designation of the species and this record appears to be an error based on the conflation of *Aphanotrigonum* with *Bradysia macfarlanei* (Jones) (Diptera: Sciaridae), another known inhabitant of *Sarracenia* pitchers in the southeastern United States. There is, obviously, no evidence of a sister group relationship between the western and eastern pitcher plant Chloropidae, given that they belong to different genera.

### *Tricimba* sp.

Material examined. USA: California: Del Norte Co., Gasquet, 31.iii.1986, emerged 14.iv.1986, D.W. Nielsen, ex larvae coll. in pitchers of *Darlingtonia californica* (1M, USNM); same data except emerged 23.iv.1986 (1F, USNM).

Remarks. Two specimens identified by *Nielsen (1990)* as an undescribed species of *Nartshukiella* (= *Tricimba*) nr. *melancholica* (Becker) were reared from a single *Darlingtonia californica* pitcher (of a total of 124 pitchers sampled) in Del Norte County, California. At the time of collection, flies were sent to C.W. Sabrosky (USNM) for identification and are still deposited in USNM. The specimens are a typical *Tricimba melancholica* group species. The tarsi are not modified as in *T. wheeleri* (Fig. 6). Given that only two specimens were reared in the course of a large-scale study of *Darlingtonia* insect associates, this may have been an opportunistic colonization of a single pitcher plant by a generalist saprophagous species. The association should be considered facultative, and more research into the natural history of this as yet undescribed species is necessary.

## CONCLUSION

There are two species, *Aphanotrigonum darlingtoniae* and *Tricimba wheeleri* (described here), which live in the pitchers of North American pitcher plants and have apparently become active partners in a fascinating ecosystem. *Aphanotrigonum* is a widespread genus with a Holarctic, New Zealand and Oriental distribution (*Nartshuk, 2012*). It is less common than *Tricimba,* which is a species-rich genus present in all biogeographical regions, except Antarctica (*Nartshuk, 2012*). Members of these genera are mostly saprophagous although there are a few predatory and parasitic species in *Tricimba* (*Nartshuk, 2014*). It is unknown whether any of the other species of *Aphanotrigonum* or *Tricimba* are closely associated with any particular plant or animal species like *A. darlingtoniae* is to *Darlingtonia californica* and *T. wheeleri* is to *Sarracenia* pitcher plants. Revisions of both genera, including their ecological habits and phylogenies, would help understand the evolution of specialization in these two species associated with pitcher plants and achieve better appreciation of chloropid diversity.

## ACKNOWLEDGEMENTS

Fieldwork in California was facilitated by Steve Gaimari and Peter Kerr (California Department of Food and Agriculture). Rob Naczi (New York Botanical Garden) provided type specimens and ecological information on *Tricimba wheeleri.* Aaron Ellison (Harvard Forest) connected us with the FM Jones archives and Steve Marshall (University of Guelph) alerted us to the *Szerlip (1975)* paper. Norm Woodley (USNM) arranged access to the USNM collection. Staff at the Biodiversity Institute of Ontario, especially Valérie Lévesque-Beaudin, collaborated on the DNA barcoding of *A. darlingtoniae.* Stephen Heard made helpful comments on the manuscript.

### Funding

This research was funded by a Natural Sciences and Engineering Research Council of Canada Discovery Grant to TA Wheeler. The funders had no role in study design, data collection and analysis, decision to publish, or preparation of the manuscript.

### Grant Disclosures

The following grant information was disclosed by the authors:
Natural Sciences and Engineering Research Council of Canada Discovery Grant.

### Competing Interests

The authors declare there are no competing interests.

### Author Contributions

- Julia J. Mlynarek conceived and designed the experiments, performed the experiments, analyzed the data, contributed reagents/materials/analysis tools, prepared figures and/or tables, authored or reviewed drafts of the paper, approved the final draft.

- Terry A. Wheeler conceived and designed the experiments, performed the experiments, analyzed the data, contributed reagents/materials/analysis tools.

## DNA Deposition

The following information was supplied regarding the deposition of DNA sequences:

The Amphanotrigonum darlingtoniae sequences described here are accessible via BOLD accession numbers: CCDB-21328-A12/LYMAA1247-14; CCDB-21328-B01/LYMAA1248-14.

## Data Availability

Specimens studied are deposited in the following collections: Lyman Entomological Museum, McGill University, Ste-Anne-de-Bellevue, Quebec, Canada (LEMQ); Personal collection of Robert F.C. Naczi (RFCN); United States National Museum of Natural History, Smithsonian Institution, Washington, D.C., USA (USNM).

## New Species Registration

The following information was supplied regarding the registration of a newly described species:

Publication: LSID: urn:lsid:zoobank.org:pub:67933A14-78D1-4A3B-AC17-B00951574F39;

Species name: *Tricimba wheeleri* urn:lsid:zoobank.org:act:C9CC48AD-AE7A-49DB-B849-2C64B6C5E812.

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
