# Peer review of "Chloropid flies (Diptera, Chloropidae) associated with pitcher plants in North America"

_PeerJ, doi:10.7717/peerj.4491_

## Round 0.1 · original submission · Minor Revisions

Please incorporate the recommended revisions or explain why they were not incorporated.

·

Basic reporting

no comment

Experimental design

no comment

Validity of the findings

no comment

Additional comments

This paper is well illustrated, problem well defined and researched. Most of my comments and corrections are added to the pdf version of the ms. In addition, I have the following points:

1. Jones (1916) also described the immature stages of Aphanotrigonum darlingtoniae, but this was not mentioned in the ms. This should be added. Also, the immature stages were reviewed in the following reference (Johannsen, O.A. (1935) Aquatic Diptera. Part II. Orthorrhapha-Brachycera and Cyclorrhapha. Cornell University Agricultural Experiment Station Memoir 177: 1-62, pls 1-12.). This reference should also be added to the ms.

2. The link to the Jones notes could be more precise and direct. I suggest that you use the link to the "Insect Associates" page.

3. Jones (1916) actually did not designate a holotype. He states "types male and female" and lists paratypes. This actually does not constitute a holotype designation and all specimens are syntypes. The USNM type number is not unique and according to the ms, the number is present on more than one specimen. The type labels were added by curators at the USNM following publication and the holotype label is not valid (but this specimen can certainly be used for the lectotype). A lectotype designation is required and recommended here. Also did you examine the "paratypes" in the other collections, i.e., at ANSP?

Reviewer 2 ·

Basic reporting

Overall a very well-written and informative piece of research. Excellent coverage of background information and context for natural history of species discussed and described. One minor suggestion would be to include a brief discussion of the known diversity and geographic range of the 2 genera containing pitcher plant associated species of Chloropidae, simply to provide further context for the rarity of this associative life history and potential inspiration for field naturalists to watch for other associations in their travels.

Experimental design

Well defined and articulated. Article and new species are registered with the ICZN Zoobank database, and LSIDs are provided along with PeerJ's required text on digital publication of new names.

Validity of the findings

No comment.

Additional comments

A well-written and engaging piece of research, and a touching tribute to a Canadian leader in Dipterology and taxonomic entomology. A few minor typos or suggestions for grammatical refinements are provided as comments in the attached PDF, but otherwise a very clean and clear manuscript.

Annotated reviews are not available for download in order to protect the identity of reviewers who chose to remain anonymous.

·

Basic reporting

This paper is written in clear English, it is well referenced and relevant. The figures are relevant and of high quality. This is a description of a new species, therefore no raw data exists.

The BOLD data for the two described specimens of Aphanotrigonum darlingtoniae could not be accessed as it is not publicly available. However, it seems to be uploaded to BOLD, but I am not an expert in this system.

I could not find the new species data in Zoobank, but I am not an expert on this system and might have done something wrong.

Experimental design

It is a new species, therefore fills a knowledge gap. The author has investigated the subject to a high standard and the methods are appropriate and can be replicated. The information provided is sufficient.

Validity of the findings

The conclusions are well stated and the paper is of a high standard.

Additional comments

This paper includes the description of one new species and it is very useful that a comprehensive re-description of a second species is included. It is a valuable contribution to the knowledge of Chloropidae associated with pitcher plants.
I could not find the specimens in BOLD or zoobank and I suggest to correct this, but I am not an expert in these systems, so might have done something wrong.
I found some small typos in the ms as follows:
line 181 - land should be and
line 182 - ines should be lines
line 186 - pollnose should be pollinose

---

## Round 0.2 · accepted · Accept

Thank you for your detailed letter describing the changes made.